# Psychosocial, cultural, and academic challenges to Saudi Arabian students in Australia

Tahir Jameel[1]*, Mukhtiar Baig[2], Saba Tariq[3], Zohair Jamil Gazzaz[1], Nadeem Shafique Butt[4], Nouf Khaleel Althagafi[5], Eman Yahya Hazazi[5], Razan Saleh Alsayed[5]

**1** Department of Internal Medicine, Faculty of Medicine Rabigh, King Abdulaziz University, Jeddah, KSA, **2** Department of Clinical Biochemistry, Faculty of Medicine Rabigh, King Abdulaziz University, Jeddah, KSA, **3** Department of Pharmacology, University Medical & Dental College, University of Faisalabad, Faisalabad, Pakistan, **4** Department of Family & Community Medicine, Faculty of Medicine Rabigh, King Abdulaziz University, Jeddah, KSA, **5** Fifth year MBBS student, Faculty of Medicine Rabigh, King Abdulaziz University, Jeddah, KSA

* tjahmed@kau.edu.sa

**Data Availability Statement:** All data has been given the tables and figures.

**Funding:** This study project (233394) was funded by the Deanship of Scientific Research (DSR), KAU, Jeddah, under grant number (G:667-828-

## Abstract

### Objectives

This study investigated the perceptions of Saudi Arabian medical disciplines students undergoing training in various institutes of Australia regarding psychosocial, cultural, and academic challenges.

### Methods

This cross-sectional study was from March 15 to June 15, 2019. Data were collected by an online questionnaire. It consisted of questions regarding demographic, psychosocial, cultural, and academic challenges. Two hundred nineteen students studying in Australia responded to our questionnaire.

### Results

Of the total 219 students, 13(6.0%) were undergraduate, 167(76%) were postgraduate, and 39(18%) were Ph.D. students. For most students (171[79.2%]), Australia was the country of choice for studying. Most of them were satisfied with their academic performance and adjustment to the Australian way of living. Most of the students (180[82.2%]) showed satisfaction over the availability of fair chances of their religious practices in Australia. Few of them faced difficulties coping with the Australian climate (25[11.4%]), homesickness (59[26.9%]), and food and dietary sources (44[20.1%]). Students were overall satisfied with the student advisory system (156[71.2%]), university assessments (147[67.2%]), and available research facilities (170[77.6%]). Among participants, 77 (35.1%), 119(54.3%), and 23(10.5%) students indicated that they wished to stay in Australia only until completion of their studies, temporarily and permanently, respectively.

1441). The authors wish to thank DSR, KAU, for the technical and financial support of this project. The funders had no role in study design, data collection and analysis, decision to publish, or preparation of the manuscript.

**Competing interests:** The authors have declared that no competing interests exist.

## Conclusions

Our findings showed that Saudi students in Australia had strong psychosocial well-being, cultural integration, and academic success. Most of them were satisfied and adjusted well to Australian culture.

## Introduction

It has been an ancient tradition to travel to far-off countries to acquire knowledge. A lot of students travel across the world to acquire higher degrees. Similarly, medical students attend reputed teaching institutions throughout the world [1]. The Ministry of Education of the Kingdom of Saudi Arabia (KSA) sponsored 174333 students for education abroad in 2017 [2]. The figure shows the commitment and sincerity of the government to raise the number of learned citizens. When these students return after attaining higher qualifications, they assist in upgrading the academic standards in the Kingdom [3].

It has always been challenging for international students to settle down in a country with a different language and culture from their home country, especially when the educational set up is unfamiliar [4]. It is a difficult start when one is not accustomed to the norms of society. Moreover, international students may not understand the lectures and home assignments when there is a language barrier. In the case of interactive lectures, it becomes difficult for them to participate in active discussions and question answer-sessions in classes and ward rounds due to the language barrier, which may lead to lack of participation [5]. International students, especially medical undergraduates, and postgraduates are among the most successful students in their home countries after getting training from abroad [6]. This could be because of the better training opportunities and advanced technology available at the developed countries universities.

With the rapidly changing geopolitical circumstances, it is not easy to face religious challenges, especially for the Arab Muslim students, because, at times, some students on campus become very critical of specific religious beliefs [7]. It also becomes challenging to complete the assignments allotted to students with equal representation of female students where interaction is quite open and needs fluency in English [8]. However, most Saudi students are presently familiar with the importance of the English language, which has been adopted as a teaching language in most universities at home [9]. Students are supposed to attend interactive language courses before joining the international teaching institutes. Now they face fewer problems than their senior students who joined international institutes years back without proper English language coaching [10].

Australia is an open and tolerant society and much more welcoming than many western countries. International students, especially from Saudi Arabian backgrounds, may feel difficulties while adapting to a rather open and different setup than their home country. Some of the factors may be the difference in teaching and learning approaches. In the KSA, there is gender segregation in teaching institutes. Male & female students' study in separate schools and institutions. While in Australia, there is coeducation, and emphasis is on open discussions among students, and teachers act as facilitators. Other differences being the mixed gathering and social activities [8]. Non-participation in social activities leads to loneliness among international students, especially those from Muslim cultures [5, 11]. All the religions are being practiced in Australia quite peacefully [12]. The students admitted to various medical universities and teaching institutes are very comfortable with the teaching environment and society's acceptance [13]. Most Saudi students acquiring postgraduate qualifications are residing with their families, and their children's exposure to Australian society is encouraging [14, 15]. This

study investigated the perceptions of Saudi Arabian medical disciplines students undergoing training in various institutes of Australia regarding psychosocial, cultural, and academic challenges during their stay.

## Methods

### Study design and setting

This cross-sectional study was conducted from March 15 to June 15, 2019. Ethical approval was taken from the Ethical Review Committee of the University of Faisalabad, Pakistan (Reference No. TUF/Dean/2020/66). The data collection protocol adhered to institutional and national ethical standards, as well as the Helsinki Declaration. Data anonymity and confidentiality are preserved.

The data were collected using the snowball sampling method. Raosoft sample size calculator was used to calculate the representative sample size. The calculated sample size was 208 at a margin of error of 5% with a confidence level of 95%, response rate 50%, and population size 450 (Students' clubs in Australia confirmed that 450 Saudi students were registered for an undergraduate medical degree, postgraduate and Ph.D. in medical sciences at various institutes during the study period).

### Data collection procedure

We invited only those Saudi students registered for an undergraduate medical degree, medical graduates at various institutes in Australia for postgraduation, and Ph.D. in different medical sciences disciplines. We were in touch with the cultural attaché in the Royal Saudi Arabian Embassy, Canberra, to get the contact details of various Saudi Arabian student clubs in different Australian institutes having medical teaching facilities. We received a good response from various student clubs and got the contact number of many medical students. The questionnaire's link was sent through electronic media (WhatsApp, Facebook messenger, and through emails) & requested those students to help us reach other medical students (snowball technique). Our student co-researchers also helped us to reach many Saudi students and 50 students were directly contacted. Of the total 450, 219 students responded to our questionnaire, so the response rate was 49%. It is less likely that the composition of our respondents was different from the non-respondents because, in our study, there were respondents from both genders, married and unmarried, and all three groups (undergraduate medical degree, medical graduates doing postgraduation, and Ph.D. in different medical sciences disciplines).

### Data collection instrument

An online questionnaire was formulated using the LimeSurvey Service to collect the participants' experiences, observations, and difficulties during their stay in Australia. We gave a brief statement about the research objectives and requested their participation at the start of the questionnaire. The completion of the online questionnaire was considered the students' consent to participate in the study.

The questionnaire was formulated in English with the help of already published studies [13, 15]. We found that these questions thoroughly evaluate the psychosocial, cultural, and academic challenges Saudi Arabian students face in Australia. Therefore, these questions were used in the present study. The content and construct validity were ensured by two senior professors and a medical educationist proficient in English by evaluating these questions. Moreover, the questionnaire was sent to a few students to check its comprehension and later modified after receiving content experts' and students' suggestions. The questionnaire

reliability was calculated by Cronbach's alpha, and its value was 0.93. It consisted of the demographic section with the questions regarding age, gender, site of education in Australia, name of the institution & city, name of the faculty, level of study (undergraduate, postgraduate, or Ph. D.), marital status, accommodation, city of residence in KSA, residing with family or without family, length of stay in Australia, number of visits to KSA since the start of education & overall financial condition. The section on academic satisfaction contained the questions regarding former education, academic performance in Australia, satisfaction regarding various academic activities, including group discussions in mixed gatherings. The psychosocial part was concerned with the knowledge & impression about Australia before their arrival, any experience regarding prejudice treatment in Australia. The questions were also directed to find out adjustments in the Australian way of living and climate, attitudes of fellow Australians, interaction with students of the opposite sex in day-to-day dealing, and group discussions. Moreover, we also explored any difficulty getting temporary jobs during studies, the extent of homesickness, adjustment with the environment, various foods, schooling of children, and others.

We also asked about the children mixing up with Australian friends, family visits to Australian friends' homes, participation in Australian functions, and traditional get-together. Students' satisfaction was measured on a five-point Likert scale [16]. If more than 60% of students selected options highly satisfactory and satisfactory or unsatisfactory and highly unsatisfactory, were considered satisfied and unsatisfied, respectively.

**Statistical analysis.** Collected data was evaluated on SPSS-26. Frequencies and percentages for the different variables were determined [17]. Two-proportion and chi-square testing were conducted to investigate the comparison between the different variables. All the p—values <0.05 have been considered significant.

## Results

Two hundred and nineteen students [36(16.4%) females and 183(83.6%) males] participated in this study, and the data were obtained from 42 Australian institutes. The general characteristics of the study participants are shown in Table 1. There were 13 (6.0%) undergraduate, 167 (76%) postgraduate, and 39 (18%) Ph.D. students. For most of the students,171(79.2%), Australia was the country of choice for studying. More than two-thirds of students, 154(69.5%), had travelled to KSA one or more than once after arriving in Australia. The majority of the students, 131(59.85%), were financed by the Saudi government, followed by university scholarships, self-financed, and other resources (Table 1).

The comparison of satisfaction score by study variables showed that males were more satisfied than females (p = 0.002). Students whose spouses were also studying in Australia were more satisfied than students without spouses (p = 0.004). Other variable differences are given in Table 2.

In two-thirds of the students,172(78.6%), Australia's financial status was highly satisfactory and satisfactory. Most of the students' impressions, 166 (75.8%), were highly satisfactory and satisfactory about Australia and its people. Four out of five students were highly satisfied and satisfied with their academic performance and studies and adjusted to the Australian way of living. Most of the students, 180(82.2%), were highly satisfied and satisfied with the availability of fair chances of their religious practices and facilities in Australia. A few of the problems Saudi medical disciplines related students faced in Australia were coping with the Australian climate (25[11.4%]), homesickness (59[26.9%]), food and dietary sources (44[20.1%]). Students were overall satisfied with the student advisory system 156(71.2%), university assessments 147(67.2%), available research facilities 170(77.6%), and speaking English at home with family 133(60.7%) (Table 3).

**Table 1. General characteristics of study participants and their choice of study, number of visits to Saudi Arabia, and study financer.**

| Variables | | n | (%) |
|---|---|---|---|
| Gender | Female | 36 | (16.4) |
| | Male | 183 | (83.6) |
| Age in years | < 30 | 136 | (62.1) |
| | >30 | 83 | (37.9) |
| Residing at | Hostel | 27 | (12.3) |
| | Self-rented home | 179 | (81.7) |
| | Cultural building | 13 | (5.9) |
| Marital status | Married | 153 | (69.9) |
| | Unmarried | 66 | (30.1) |
| Spouse is also a student in Australia | Yes | 13 | (8.4) |
| | No | 140 | (91.6) |
| If married, number of kids | 0 | 18 | (13.0) |
| | 1 | 39 | (25.0) |
| | 2 or more | 110 | (72.0) |
| Residing with family | Yes | 138 | (63.6) |
| | No | 79 | (36.4) |
| Study level | Undergraduate | 13 | (5.9) |
| | Postgraduate | 167 | (76.3) |
| | PhD | 39 | (17.8) |
| Completed course/s in Australia | Bachelor | 19 | (8.7) |
| | Master | 46 | (21.0) |
| | Other courses | 18 | (8.2) |
| | No course completed | 136 | (62.1) |
| Australia was the country of choice for studying | Yes | 171 | (79.2) |
| | No | 45 | (20.8) |
| Number of visits to KSA after arrival in Australia | 0 | 65 | (30.5) |
| | 1 | 61 | (28.6) |
| | 2 | 34 | (16.0) |
| | 3 | 15 | (7.0) |
| | 4 | 19 | (8.9) |
| | > = 5 | 19 | (8.9) |
| Financer of your Study | Self-financed | 21 | (9.6) |
| | Saudi Govt | 131 | (59.8) |
| | University scholarship | 42 | (19.2) |
| | Other | 25 | (11.4) |

Students' responses were varied about getting work while studying in Australia; 97(44.3%) stated it was difficult to find, and 87(39.7%) responded they never tried (Fig 1).

Among participants, 77 (35.1%), 119(54.3%), and 23(10.5%) students indicated that they wished to stay in Australia only until completion of their studies, temporarily and permanently, respectively (Fig 2).

## Discussion

The current literature indicates that this is the first study regarding Saudi medical disciplines related students' experiences in Australia who were registered for an undergraduate medical degree, postgraduate, and Ph.D. in medical sciences at various institutes. A definite cultural

**Table 2. Comparison of satisfaction score by study variables.**

| Variables | | Median (Q1-Q3) | p-value |
|---|---|---|---|
| Gender | Male (n = 183) | 3.90 (3.66–4.11) | 0.002 |
| | Female (n = 36) | 3.78 (3.53–4.03) | |
| Age | <30 (n = 136) | 3.78 (3.53–4.03) | 0.910 |
| | >30 (n = 83) | 3.97 (3.58–4.18) | |
| Residing | Hostel (n = 27) | 3.80 (3.34–4.00) | 0.433 |
| | Self-rented home (n = 179) | 3.90 (3.61–4.10) | |
| | Cultural Building (n = 13) | 3.83 (3.57–4.23) | |
| Marital status | Married (n = 153) | 3.87 (3.56–4.10) | 0.493 |
| | Unmarried (n = 66) | 3.87 (3.60–4.11) | |
| Level of study | Undergraduate (13) | 3.83 (3.53–4.07) | .207 |
| | Postgraduate (167) | 3.83 (3.63–4.03) | |
| | PhD (39) | 3.97 (3.57–4.17) | |
| Financer of study | Self-financed (21) | 3.93 (3.51–4.34) | .020 |
| | Saudi Govt (131) | 3.90 (3.67–4.10) | |
| | University scholarship (42) | 3.73 (3.52–3.97) | |
| | Other (25) | 3.73 (3.49–4.00) | |
| Residing with family | Yes (138) | 3.88 (3.56–4.10) | 0.928 |
| | No (79) | 3.83 (3.60–4.09) | |
| Spouse is also studying in Australia | Yes (13) | 3.90 (3.57–4.13) | 0.004 |
| | No (140) | 3.83 (3.52–4.03) | |
| Any sign of prejudice you faced | Slight (21) (9.5%) | 3.80 (3.47–4.04) | 0.003 |
| | No (198) (90.5%) | 3.90 (3.63–4.10) | |

difference exists between KSA and Australia. The most challenging experience for Saudi male and female students is the mental adjustment in a different environment [18]. Australian universities are well accustomed to absorbing international students by easing out many difficult adjustment stages. The newly arrived student's initial challenge is adjusting to a society where language, social culture, and day-to-day dealing differ from one's parent society. In university life, one is expected to speak out his point of view from the very start of academic activities, and then one finds a diverse society in which expressing oneself in discussion and conversation is essential. [12, 15]. Saudi Arabian students attend English learning classes during summer vacations, so the language barrier gets diluted due to better education standards in local institutions [12].

Two-third of the study participants were married and living in rented accommodations. Most of the study participants were married because most of them were postgraduate students or pursuing a Ph.D. In comparison to our findings, a previous study found that Saudi students were homesick because they couldn't keep their families with them [19]. Those male Saudi students were undergoing training for ELICOS (English Language Intensive Course for Overseas Students) in an Australian University. Later Saudi Arabian cabinet passed a law that international students could keep their spouses with them during training abroad by special monthly spouse allowance [20]. As per responses recorded by our participants, they were socially active and didn't feel any hesitation while contacting the opposite-sex students and their families intermixed with Australian friends.

## Psychosocial aspects

International students face multiple psychosocial and stress experiences during their stay. They are primarily academic pressures, language difficulties, developing a social relationship

**Table 3. Students' responses to various questions on a five-point scale.**

| Questions/Statements | Highly satisfactory | Satisfactory | Uncertain | Unsatisfactory | Highly unsatisfactory |
|---|---|---|---|---|---|
| | n(%) | n(%) | n(%) | n(%) | n(%) |
| How is your financial situation here? | 38(17.4) | 134(61.2) | 25(11.4) | 17(7.8) | 5(2.3) |
| How well do you feel former education helped you in your study in Australia? | 29(13.2) | 126(57.5) | 32(14.6) | (10.0) | 10(4.6) |
| How much satisfied were you about your knowledge regarding Australia before coming here? | 32(14.6) | 108(49.3) | 49(22.4) | 23(10.5) | 7(3.2) |
| How was your impression of Australia and its people when you arrived here? | 53(24.2) | 113(51.6) | 27(12.3) | 20(9.1) | 6(2.7) |
| How well are you performing academically? | 40(18.3) | 133(60.7) | 33(15.1) | 11(5.0) | 2(.9) |
| Have you got a fair chance to practice your religious routines here? | 71(32.4) | 109(49.8) | 28(12.8) | 9(4.1) | 2(.9) |
| How much facilities are available to you for practicing your religion? | 44(20.1) | 132(60.3) | 28(12.8) | 12(5.5) | 3(1.4) |
| To what extent you are satisfied with your studies in Australia. | 52(23.7) | 121(55.3) | 21(9.6) | 13(5.9) | 12(5.5) |
| To what extent you are satisfied with academic group discussions. | 38(17.4) | 107(48.9) | 40(18.3) | 25(11.4) | 9(4.1) |
| To what extent you are satisfied with understanding written English. | 50(22.8) | 113(51.6) | 27(12.3) | 22(10.0) | 7(3.2) |
| To what extent you are satisfied with reading Journals, books etc. | 46(21.0) | 107(48.9) | 40(18.3) | 19(8.7) | 7(3.2) |
| To what extent you are satisfied with progress in your studies. | 50(22.8) | 122(55.7) | 30(13.7) | 11(5.0) | 6(2.7) |
| How do you assess your relationship with your supervisor or teachers? | 44(20.1) | 115(52.5) | 38(17.4) | 16(7.3) | 6(2.7) |
| To what extent you are satisfied with adjusting to Australian way of living. | 56(25.6) | 119(54.3) | 25(11.4) | 13(5.9) | 6(2.7) |
| To what extent you have coped with Australian climate. | 54(24.7) | 108(49.3) | 32(14.6) | 18(8.2) | 7(3.2) |
| To what extent you have coped with problems regarding homesickness. | 27(12.3) | 95(43.4) | 38(17.4) | 37(16.9) | 22(10.0) |
| To what extent you are satisfied with food and dietary sources in Australia. | 44(20.1) | 98(44.7) | 33(15.1) | 31(14.2) | 13(5.9) |
| How do you feel regarding your day-to-day activities? | 36(16.4) | 128(58.4) | 34(15.5) | 15(6.8) | 6(2.7) |
| How satisfied are you with the way Australians treat you? | 50(22.8) | 119(54.3) | 31(14.2) | 15(6.8) | 4(1.8) |
| How much satisfied are you in interaction with your fellow Australians? | 40(18.3) | 111(50.7) | 49(22.4) | 14(6.4) | 5(2.3) |
| If you have children, are you satisfied with their education here? | 42(19.2) | 97(44.3) | 60(27.4) | 13(5.9) | 7(3.2) |
| How satisfied are you with your academic activities? | 31(14.2) | 130(59.4) | 38(17.4) | 16(7.3) | 4(1.8) |
| How satisfied are you with the student advisory system? | 37(16.9) | 119(54.3) | 40(18.3) | 16(7.3) | 7(3.2) |
| How satisfied are you with University assessments? | 31(14.2) | 116(53.0) | 38(17.4) | 24(11.0) | 10(4.6) |
| How satisfied are you with available research facilities? | 52(23.7) | 118(53.9) | 29(13.2) | 15(6.8) | 5(2.3) |
| How satisfied are you with the Australian style of clothing? | 44(20.1) | 105(47.9) | 43(19.6) | 18(8.2) | 9(4.1) |
| How satisfied are you with your interaction with students of opposite sex? | 43(19.6) | 118(53.9) | 39(17.8) | 13(5.9) | 6(2.7) |
| How do you feel about your interaction with opposite-sex students during group discussions? | 44(20.1) | 118(53.9) | 34(15.5) | 15(6.8) | 8(3.7) |
| To what extent you are satisfied with speaking English at home with your family. | 35(16.0) | 98(44.7) | 39(17.8) | 32(14.6) | 15(6.8) |

with fellow students, and teaching staff [21]. Alsahafi & Shin pointed out that KSA has been among the top ten sources of international students in Australia since 2011 [12]. It has been pointed out that the few significant challenges for Saudi students in the educational environment were open discussions among male and female students and the completion of group assignments with female colleagues [22].

About three out of four of our participants were satisfied with their ability to cope with fellow Australian students and in day-to-day dealing with the locals. Several recent studies have suggested that by visualizing the better living facilities, quite a proportion of international students prefer to settle down in Australia by acquiring citizenship [23, 24]. While among our study participants, only a small proportion (10%) mentioned their wish to stay in Australia permanently after completing their studies.

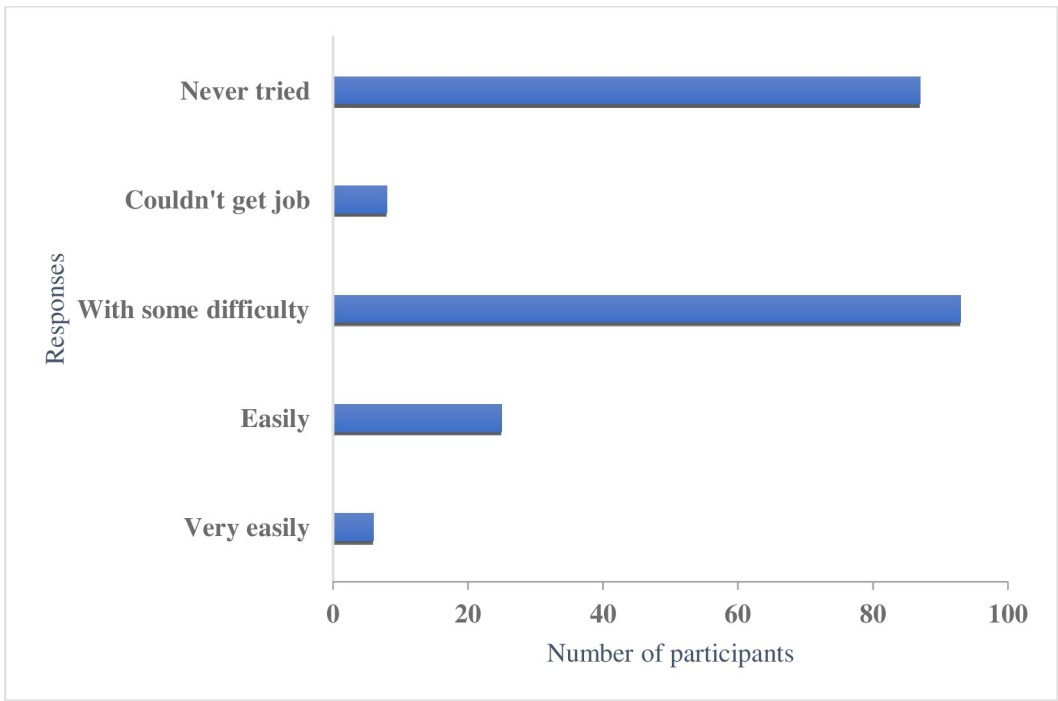

**Fig 1. Students' responses regarding getting work while studying in Australia.**

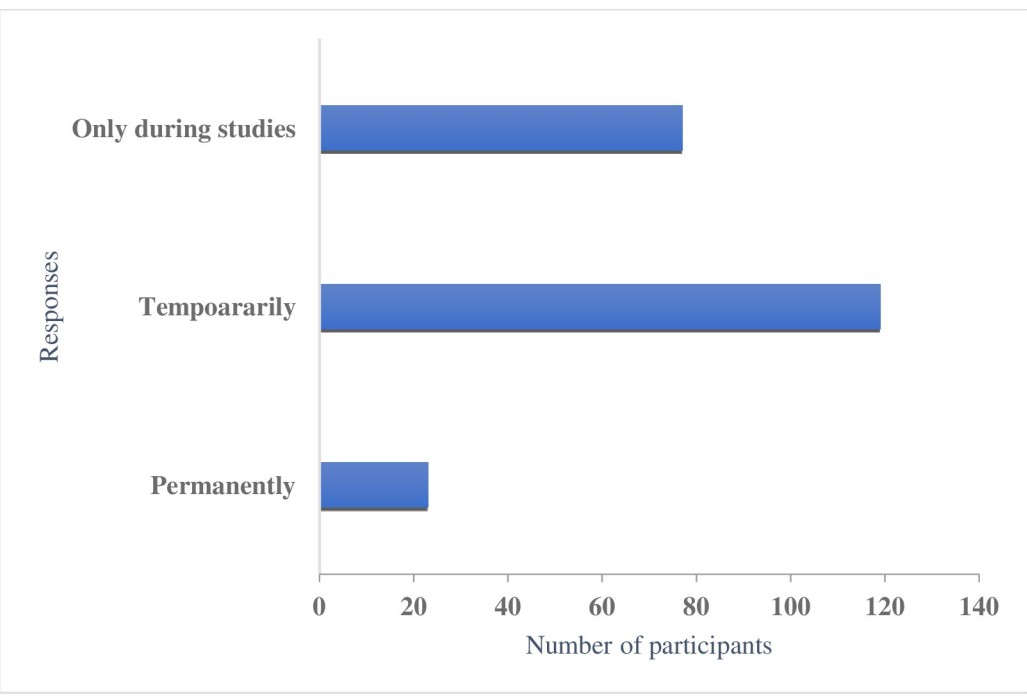

**Fig 2. Students' responses regarding staying in Australia after the completion of their studies.**

## Cultural aspects

The cultural atmosphere in KSA is religious, so naturally, the Muslim students look for a proper place and atmosphere for religious activities like facilities for prayers during working hours. Australian teaching institutions are supposed to provide a prayer room for their Muslim students [25]. A recent study mentioned that Australia being open, fair, and tolerant of all cultural activities, including all religions, is a favorite place for most international students, including the female students from KSA [26]. The majority of study participants were satisfied with the fair provision of facilities for their religious practices on their campuses and around the residences. A large majority of our study participants (90%) denied any discriminatory practice on the campus or in society. Therefore, most of our participants mentioned their free intermixing with Australian friends and participated in different cultural activities in their vicinity. Australian society is relatively tolerant, and international students don't face any attitude or discriminatory behavior; moreover, the teaching institutions are very well reputed [27]. This is why most participants mentioned Australia as their first choice for professional training.

## Financial aspects

A previous study mentioned that because of lack of finances and cultural restrictions, most male international students couldn't take their families along with them and be distracted from academic activities [19]. However, that study was published in 2009, and now things have been changed. In our study, the complaints of homesickness were common in unmarried students and those missing the activities of combined family gatherings. Thirty-six students in our study were females, and thirteen out of these 36 went to Australia as spouses of the male students but got admission in undergraduate medicine classes. This clearly shows our participants' stable financial status and their husbands' encouraging attitude allowing them to get admitted to medical schools. The majority of Saudi students declared their financial status as satisfactory / highly satisfactory. A study pointed out that financial issues are one of the major causes of disturbance for international students because running & unexpected living charges disturb the students and sometimes, they need to get psychiatric consultations [28]. Australian universities have arranged for part-time jobs for international students to overcome any financial constraints [29]. But our participants' financial status was stable, so 40% of our participants mentioned that they have never tried for any part-time job. Part-time jobs can be easily obtained in smaller cities & towns compared to big cities [30], and 14% of our study participants pointed out that they got it without difficulty when required a part-time job. Ninety percent of the students committed that they have been financed by the KSA government, different university scholarships, or NGOs. Only a small percentage of the study cohort mentioned that their families are bearing their educational expenses. Most of these students were spouses who decided to get enrolled in undergraduate medical training.

## Academic aspects

Several studies have pointed out that learning and mastering English is a key to achieving academic success in any international institute. Most international students faced many problems in classroom discussions, completing the assignments, and various academic activities [18, 31, 32]. Realizing the global demand for the English language in all fields, especially in science, KSA introduced English as a compulsory subject from early school teaching and announced it as the official teaching language in all universities [33]. Saudi students are motivated to take English learning classes during their spare time, such as summer vacations [34, 35]. The preparatory year was introduced in KSA at the start of undergraduate medical training. Students get

the English language training and other basic science subjects throughout the year with the concept that they would be grasping the medical subject effectively [30, 36]. Arabic is a powerful language, and almost all Saudi citizens prefer to speak in Arabic. Time would be required for the English language to be the sole language during academic sessions and discussions. Our study cohort included students from various medical disciplines. About 75% of our participants felt very confident in understanding written and verbal English. Most of our participants admitted that KSA's former education helped them a lot in their studies in Australia. They didn't complain of any problems during their classroom discussions and completing the group assignments. Overall, four out of five study participants were either satisfied or highly satisfied with their academic performance in their studies and adjusted to the Australian way of life. The majority expressed confidence in university advisory systems for the students, university assessments, and research facilities.

Interestingly, many of the participants mentioned they speak English at home with their families. Among married students, most students had one or more children in Australia. They showed satisfaction regarding their children's schooling and accepted that they participate in various social activities with Australian families. Most universities keenly socialize their international students to achieve better academic records [18]. Similar to our study, an Australian study explored the Saudi nurses' experiences of studying in Australia and mentioned that students perceived their learning has a transformative effect on their professional and psychological growth [37].

## Implications of the study

KSA is among the top 10 countries sending its youth to world-known institutions for higher education [38]. Most Saudi students prefer the USA, UK, and Canada for postgraduate studies [39]. The present study has highlighted Australia's conducive and friendly environment where Saudi Muslim students of both genders felt comfortable in a friendly and welcoming society. Our results showed that male and female students expressed satisfaction regarding all aspects of their academic commitments except a few odd experiences, which are part of day-to-day life. We hope the study results would be beneficial not only for students to choose the right destination but also to provide a guideline for the selectors.

## Future recommendations

Saudi students going abroad for higher studies are mostly well prepared, as evident from our results. There is always a need for improvement, and we recommend the following measures to be ensured before moving to overseas teaching institutions.

1. English proficiency must be ensured, and regular sessions must be arranged for mutual discussions in English. Mostly in Saudi schools and colleges, teacher-student interaction is in the native Arabic language. Therefore, we recommend teachers ensure that the discussion medium should be only English during classroom discussions and explanations. Students must be trained for the specific English dialect of the area where they are supposed to land.

2. Students must be briefed regarding the destination country's culture, history, and social setup. They should be mentally prepared before landing and don't feel difficulty mixing up with other students and social gatherings. It is critical for international students to be confident and be able to socialize regarding their academic schedules to fit in the culture of the institutions and society.

3. Students must develop the habit of book reading as in higher studies, one must complete the assignments quickly and may require lots of studies from various sources to compile the assignment.

4. Saudi cultural centers in Embassy must be having track & active liaison with their students in various institutions so that the help may be provided in any unforeseen situation.

## Limitations of the study

This qualitative study's findings are based on the data collected from a purposefully selected group of students enrolled at various Australian institutes. The findings may not be generalized as larger mixed studies may be required to access the Saudi students' experiences in different disciplines across the Australian Universities. Another important limitation was the online nature of the study. Online surveys have several drawbacks, including poor response rate, and it is hard to ascertain whether the right person completed the questionnaire. It's difficult to contact people who don't use social media, and frequent requests to finish the survey can be frustrating for participants and often backfire [40]. A subjective self-reported measure and accompanying biases such as acquiescence and dissent biases are also possible.

## Conclusions

Our findings showed that Saudi students in Australia had strong psychosocial well-being, cultural integration, and academic success. Most of our students coped up with the difficulties quite successfully. Most of them were satisfied and adjusted well to Australian culture.

## Acknowledgments

We are thankful to our students Taha Mohammed Aljifri, Bander Abdulaziz Almaghrabi, Mohammed Ibrahim Albejad, Abdulakhaliq Aali Alghamdi, and Abdulaziz Naif Kadasa for their help in data collection.

## Author Contributions

**Conceptualization:** Tahir Jameel, Saba Tariq.

**Data curation:** Nadeem Shafique Butt, Nouf Khaleel Althagafi, Eman Yahya Hazazi, Razan Saleh Alsayed.

**Formal analysis:** Mukhtiar Baig, Nadeem Shafique Butt.

**Funding acquisition:** Tahir Jameel.

**Investigation:** Tahir Jameel, Eman Yahya Hazazi.

**Methodology:** Tahir Jameel, Mukhtiar Baig, Saba Tariq, Zohair Jamil Gazzaz, Nouf Khaleel Althagafi, Eman Yahya Hazazi, Razan Saleh Alsayed.

**Project administration:** Tahir Jameel, Zohair Jamil Gazzaz, Nouf Khaleel Althagafi, Razan Saleh Alsayed.

**Validation:** Nadeem Shafique Butt.

**Writing – original draft:** Saba Tariq.

**Writing – review & editing:** Mukhtiar Baig, Zohair Jamil Gazzaz.

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
