## [Decision Letter · Decision Letter 0]

18 Mar 2021

PONE-D-21-00081

Psychosocial, cultural and academic challenges to Saudi Arabian International medical students in Australia

PLOS ONE

Dear Dr. Jameel,

Thank you for submitting your manuscript to PLOS ONE. After careful consideration, we feel that it has merit but does not fully meet PLOS ONE’s publication criteria as it currently stands. Therefore, we invite you to submit a revised version of the manuscript that addresses the points raised during the review process.

We look forward to receiving your revised manuscript.

Kind regards,

Prof. Ritesh G. Menezes, M.B.B.S., M.D., Diplomate N.B.

Academic Editor

PLOS ONE

Journal Requirements:

Additional Academic Editor Comments:

• Why was ethical approval for the present study obtained from the Ethical Review Committee of the University of Faisalabad based in Pakistan while the study was funded by King Abdulaziz University based in Saudi Arabia? The first author is based in Saudi Arabia. Why wasn’t ethical approval obtained from King Abdulaziz University, Saudi Arabia?

• Keywords: Add ‘’medical student’’.

• Introduction-2nd line: Replace ‘’SA” with ‘’Saudi Arabia (SA)”. SA is used as an abbreviation for the first time in the main text here.

• Introduction-1st paragraph-Last line: There is no mention of ‘’Kingdom’’ earlier in the paragraph.

• Introduction-2nd paragraph-1st sentence: Provide a reference to support this statement at the end of the sentence.

• Introduction-2nd paragraph-5th line: Embracing or embarrassing?

• Introduction-2nd paragraph-5th line: How is it embarrassing? Provide details rather than a blanket statement.

• Introduction-2nd paragraph-Last 2 sentences: The ‘’connect……’’ is missing with the rest of the paragraph. Moreover, the last sentence lacks clarity.

• Introduction-3rd paragraph-6th line: Good universities?

• Introduction-Last paragraph-7th line: Mention who are you referring to in relation to ‘’experiences and learnt lessons’’.

• Introduction-Last paragraph-8th line: Delete ‘’was’’.

• Introduction: You have referred to the ‘’Australian culture’’ in the conclusion section of the text. Although the Australian culture is briefly touched upon in the introduction section, strengthen the introduction to the Australian culture in the introduction section.

• Methods: Specify the name of the institution from where the ethical approval was obtained.

• Methods: What do you mean by “Students’ consent was taken, and filling the questionnaire was also considered their consent to participate in the study.”? Was the first consent obtained on-line as well? How was it obtained? This sentence needs to be clarified and accordingly revised. It is not clear regarding how exactly consent was obtained.

• Methods: What do you mean by ‘’religious rights wish to stay further in Australia”?

• Methods: Provide a reference for ‘’five-point Likert scale’’.

• Methods: Provide a reference for ‘’SPSS’’.

• Methods: The methods section requires further work. Reorganize the methods section into relevant categories (example: setting/design, sample/participants, procedure/instruments, analysis, ethical approval). Provide further details as to how the questionnaire was developed. Mention further details regarding when the study was conducted. The title suggests that the participants were medical students (Saudi Arabian medical students studying in Australia), but no mention of medical students in the methods section. How were the participants recruited? How was the sample size calculated? How was the questionnaire distributed online? Provide description of your sample. Two hundred and nineteen (219) Saudi Arabian medical students studying in Australia participated in the present study (1st line of the results section). What was the total number of Saudi Arabian medical students studying in Australia at the time of conducting the survey? What was the response rate? It is important to state the response rate of the survey (219 of how many sent). Questionnaires were sent to how many (potential participants) in total. The percentage of people who responded is an important criterion to judge the statistics. Was there any pattern to those not responding? How likely is it that the composition of all the people to whom the questionnaire was sent, is significantly different from those who responded to the questionnaire?

• Results: Thirty-nine participants (39) were pursuing a PhD. Did you consider even the PhD students as medical students?

• Results-Table 1: What do you mean by ‘’completed course/s in Australia’’? Needs further elaboration.

• Results-Table 1: It would be more interesting to relate the ‘’number of visits to Saudi Arabia after arrival in Australia’’ with the duration of stay in Australia. How many days were spent in Saudi Arabia during these visits?

• Results-Financier of your study: What is the criterion to differentiate ‘’Saudi government’’ from ‘’University scholarship’’? Aren’t most of the universities in Saudi Arabia run by the government. Provide details to avoid further questions and accordingly revise the categories. I would like to know what ‘’others’’ stands for.

• Results-Table 1: Should it be ‘’other degree’’ or ‘’other courses’’ (completed course/s in Australia)?

• Results: What is the age of the participants?

• Discussion-1st sentence: Are you referring to ‘’Saudi Arabian medical students’’?

• Discussion-2nd paragraph-1st sentence: “Early marriage is part of Saudi Arabian culture, so most of our study participants (70%) were married ………. ‘’. You haven’t reported the age of the participants to begin with and therefore how could you provide this reason for most of the study participants being married. It appears to me that most of the participants are married because most of them are postgraduate students or pursuing a PhD; only 13 participants were undergraduate students.

• Discussion-2nd paragraph-Last sentence: Revise ‘’previous study by Midgley W. (2009)’’.

• Discussion-2nd paragraph-Last sentence: Who were the Saudi students in the previous study? Background/course pursuing?

• Discussion-Psychological aspects-1st paragraph: Provide a reference at the end of the 3rd sentence (4th line).

• Discussion-Cultural aspects-2nd line: State the religious activities.

• Discussion-Cultural aspects: Were questions on ‘’halal’’ food included in the questionnaire?

• Discussion-Financial aspects: NGOs is mentioned here. If I am not mistaken, there is no mention of NGOs in the results section.

• Discussion-Limitations of the study: The aim/objective of the study revolved around medical students. Isn’t it? Then why mention that your study related to only medical students. I vehemently do not agree with the related limitations of the study mentioned. Secondly, do not provide a blanket statement that the online nature of the study was a limitation. In what way was the online nature of the study a limitation. What were the limitations that you perceived/faced due to the online nature of the study?

• Discussion: Rewrite the limitations of the study.

• Discussion: Provide a separate paragraph on the implications of the study.

• Discussion: Provide a separate paragraph on recommendations/future directions.

• Conclusions: Draft a better paragraph on the conclusions of the study. Make sure that the conclusions drawn are based on your observations (data) and results while doing so.

- Please note that a recommendation of revision at this stage does not guarantee an acceptance.

- Address (authors’ reply to the comments + revised manuscript) all the reviewers’ comments in addition to the comments made by the Academic Editor.

Reviewers' comments:

Reviewer's Responses to Questions

**Comments to the Author**

1. Is the manuscript technically sound, and do the data support the conclusions?

Reviewer #1: Yes

Reviewer #2: Yes

2. Has the statistical analysis been performed appropriately and rigorously? 

Reviewer #1: Yes

Reviewer #2: Yes

3. Have the authors made all data underlying the findings in their manuscript fully available?

Reviewer #1: Yes

Reviewer #2: Yes

4. Is the manuscript presented in an intelligible fashion and written in standard English?

Reviewer #1: Yes

Reviewer #2: No

5. Review Comments to the Author

Reviewer #1: The research is well-written and focused on medical students with extensive details. The students sample is mainly postgraduate, in which we expect more fluency in English language in addition to more likely financial and social independence. I think this study does not represent undergraduate students.

There are a few grammar conflicts need to be addressed such as:

Page 2: Objectives should not be in the past tense

Page 3: "In the case of interactive lectures, the situation becomes rather embracing" What do you mean by this sentence?

Page 3: "At times, a group of students become a very critic of religious beliefs. It becomes challenging to complete the assignments allotted to students with equal representation of female students where interaction must be quite open and need fluency in English" It is not clear what is the point you are trying to make by these sentences.

Page 3: "good Universities" How could you define good and bad universities?

Page 5: "Religious rights wish to stay further in Australia" There is something missing in this phrase.

Page 7: ". Most of the 166(75.8%)" most of what?

Page 11: "According to a recent, study, " grammar

Reviewer #2: Readable paper, plenty of interesting findings and an under-researched group.

Should have appeal for healthcare researchers and those interested in the Middle East.

The short title needs to have ‘Saudi Arabian’ in it.

It is reasonably understandable on the whole, but I would recommend that a native English speaker review the next draft.

Further comment:

In Table 3, there is a ‘Statement’ column, but they are often phrased as questions. They need to be fixed one way or the other.

P. 10 – ‘There is a lot of’ (cf there are many)

p. 11 ‘Midgley W. (2009) reported Saudi students’ complained of being homesick as they couldn't keep their families with them [20].

So, do you think that the findings in your research could be a function of more family-friendly policies now? Other causes?

P. 12 (55.7%): do not need brackets here. You only need them when you use a number and you want to include the percentage as well: then you put the % in brackets.

P. 13 ‘SA introduced English as a compulsory subject from early school teaching and announced it as official teaching language in all the Universities [33]’. So, is this a change from [when]?

‘Interestingly, 60.7% of participants mentioned they speak English at home with their families’. So, is this a change as well?

These two comments need further interpretation.

Alzahrani (ref 16) does look like a PhD but it calls itself ‘Thesis submitted as a partial fulfilment to obtain the degree of Graduate Certificate in Research Methods’. Also, I cannot see a date on it. Can you clarify?

In the Discussion section, you state:

‘this is the first study regarding Medical students' experiences in Australia’, so you need to say this is the first survey of Saudi Arabian medical students … to differentiate it from, for example, Warren Midgley’s work.

Conclusion

‘Our findings revealed that Saudi medical students' psychosocial wellbeing, cultural adaptation, and academic success in Australia’

There is no main clause in this sentence – needs to be re-written. Furthermore, such a short conclusion does not capture your main findings. The paper should finish showing the ‘take home message’ for the readers.

Stylistic comments

Abbreviations: eg ‘it’s’ cf ‘it has’

Paragraphing: It becomes challenging (new paragraph)

Lack of clarity: ‘faced fewer problems than their (the) senior students’

Word choice: the present study was investigated

rather embracing = ‘overwhelming’?

in their homes (in their home country?)

‘supported these girls to get admitted to medical schools’, cf ‘women’

‘by visualizing the better living facilities’ cf ‘by seeing’?

‘children mixing-up (mixing in) with Australian friends’

Table 2: ‘kids’ (children)

‘so 40% of our participants mentioned that they have never tried for any part-time job. The part-time jobs can be easily obtained in smaller cities & towns compared to big cities [30], and 14% of our study participants pointed out that when they required a part-time job, they got it without difficulty. Less than half of the students got the job when required but with some difficulty’

This is interesting and would benefit from disentangling: so, is is without difficulty or with some difficulty?

Avoiding sexist language

‘to speak out his point of view’, cf ‘their’.

There is still some very cognate research the authors could look at, for example, a paper published in 2012 (Clerehan et al.) in the International Nursing Review on Saudi Arabian nurses' experiences of studying Masters degrees in Australia. The method is different (interviews) and it has a small sample, but the findings have some commonalities (and differences, and I wonder if that is a function of time passing).

Table 1 and 2 appear to overlap.

The two figures do not have captions and it's not clear if they are % or N.

6. PLOS authors have the option to publish the peer review history of their article (what does this mean?). If published, this will include your full peer review and any attached files.

Reviewer #1: No

Reviewer #2: **Yes: **R Clerehan

---

## [Author Response · Author response to Decision Letter 0]

9 May 2021

Editorial comments: We have incorporated all of your suggestions into our revision. They were very helpful. Thank you.

Reviewer 1: We have incorporated all of your suggestions into our revision. They were very helpful. Thank you.

Reviewer 2: We have incorporated all of your suggestions into our revision. They were very helpful. Thank you.

---

## [Decision Letter · Decision Letter 1]

24 Aug 2021

PONE-D-21-00081R1

Psychosocial, cultural, and academic challenges to Saudi Arabian students in Australia

PLOS ONE

Dear Dr. Jameel,

Thank you for submitting your manuscript to PLOS ONE. After careful consideration, we feel that it has merit but does not fully meet PLOS ONE’s publication criteria as it currently stands. Therefore, we invite you to submit a revised version of the manuscript that addresses the points raised during the review process.

Please submit your revised manuscript by 05-September-2021. Please include the following items when submitting your revised manuscript:

A 'Response to Reviewers' letter that responds to each point raised by the academic editor and reviewer(s). You should upload this letter as a separate file labeled 'Response to Reviewers'.A marked-up copy of your manuscript that highlights changes made to the original version. You should upload this as a separate file labeled 'Revised Manuscript with Track Changes'.An unmarked version of your revised paper without tracked changes. You should upload this as a separate file labeled 'Manuscript'.

We look forward to receiving your revised manuscript.

Kind regards,

Prof. Ritesh G. Menezes, M.B.B.S., M.D., Diplomate N.B.

Academic Editor

PLOS ONE

Journal Requirements:

Reviewers' comments:

Reviewer's Responses to Questions

**Comments to the Author**

1. If the authors have adequately addressed your comments raised in a previous round of review and you feel that this manuscript is now acceptable for publication, you may indicate that here to bypass the “Comments to the Author” section, enter your conflict of interest statement in the “Confidential to Editor” section, and submit your "Accept" recommendation.

Reviewer #1: All comments have been addressed

Reviewer #3: (No Response)

2. Is the manuscript technically sound, and do the data support the conclusions?

Reviewer #1: Yes

Reviewer #3: Yes

3. Has the statistical analysis been performed appropriately and rigorously? 

Reviewer #1: Yes

Reviewer #3: Yes

4. Have the authors made all data underlying the findings in their manuscript fully available?

Reviewer #1: Yes

Reviewer #3: Yes

5. Is the manuscript presented in an intelligible fashion and written in standard English?

Reviewer #1: Yes

Reviewer #3: No

6. Review Comments to the Author

Reviewer #1: (No Response)

Reviewer #3: In this study, Jamil et al. analyze the psychological, cultural and academic challenges Saudi Arabian students face while studying in Australia. Of the 219 students included in this analysis, we observe majority respondents to be postgraduates, and well-adjusted to the Australian system and culture.

This revised manuscript is significantly better, and authors have done a good job in responding to majority of prior queries. However, I do have a few minor comments:

1) The manuscript could be significantly improved by a copy editor, or having the document proof read as there are a number of stylistic errors in the text.

Abstract:

2) In the results portion of the abstract, I would suggest authors to mention numerical values instead of writing ‘most’ or ‘few’ in lines 3-6 of results. This would help ascertain the actual magnitude of students satisfied with their academic performance and adjustment in Australia.

3) Last four lines of Results in Abstract “Students’ responses regarding responded to a question, "Do you wish to stay in Australia after completion of your studies?", This should be removed to only include the results. Hence authors should instead write “ 77 (35.1%), 119(54.3%), and 23(10.5%) students indicated that they wished to stay in Australia only until completion of their studies, temporarily and permanently, respectively”

Introduction

1) Line 4: “The Ministry of Education of the Kingdom of Saudi Arabia (KSA) sponsored 174333 students for abroad education” This should be written as “The Ministry of Education of the Kingdom of Saudi Arabia (KSA) sponsored 174333 students for education abroad”.

2) As raised earlier, the line “In the the case of interactive lectures, the situation becomes rather embarrassing when it becomes difficult for them to take partparticipate in active discussions and question answer- sessions in classes and ward rounds” does not sit so well with me. It may not be embarrassing, rather difficult to partake in fruitful academic discussions due to language barrier, which may lead to lack of participation. I would suggest the authors to revise this sentence.

3) “International students, especially medical undergraduates, and postgraduates, are among the most successful students in their home countries [6]. This could be because of the better training opportunities and advanced technology available at the universities of the developed countries’ universities.”

This is a bit unclear to me. Do the authors mean developed countries such as Australia or the home country KSA? This should instead be written as international students being successful in their home institutions, who then in search of better training opportunities wish to go abroad. Right now it seems as if the authors are talking about students having better training opportunities in their home country, that’s why they are successful. If this is the case then why would they wish to go abroad? Some clarity would be good to make a strong introduction.

4) “While in Australia teaching is all mixed in Australia”. Suggest to use the word co-education here

Results

1) How many Australian institutes was the data obtained from?

Discussion

1) “The body of the lliterature , indicated” , suggest this to instead write as “current literature indicates”

2) Future recommendations point 1 “Arabic is a very spowerful language, and most of the time, even teacher- student interaction is in the native Arabic language. So, the teachers ensure that , the discussion medium should be only English during classroom discussions and explanations.” Suggest authors to revise this. Difficult to follow the sentence.

7. PLOS authors have the option to publish the peer review history of their article (what does this mean?). If published, this will include your full peer review and any attached files.

Reviewer #1: No

Reviewer #3: No

---

## [Author Response · Author response to Decision Letter 1]

20 Sep 2021

PONE-D-21-00081R1

Psychosocial, cultural, and academic challenges to Saudi Arabian students in Australia

PLOS ONE

Journal Requirements:

Reply:

We have checked the reference list and several references have been corrected. No reference has been removed or added.

Reviewers' comments:

Reviewer's Responses to Questions

Comments to the Author

1. If the authors have adequately addressed your comments raised in a previous round of review and you feel that this manuscript is now acceptable for publication, you may indicate that here to bypass the "Comments to the Author" section, enter your conflict of interest statement in the “Confidential to Editor” section, and submit your "Accept" recommendation.

Reviewer #1: All comments have been addressed

Reviewer #3: (No Response)

2. Is the manuscript technically sound, and do the data support the conclusions?

Reviewer #1: Yes

Reviewer #3: Yes

3. Has the statistical analysis been performed appropriately and rigorously?

Reviewer #1: Yes

Reviewer #3: Yes

4. Have the authors made all data underlying the findings in their manuscript fully available?

Reviewer #1: Yes

Reviewer #3: Yes

5. Is the manuscript presented in an intelligible fashion and written in standard English?

Reviewer #1: Yes

Reviewer #3: No

6. Review Comments to the Author

Reviewer #1: (No Response)

Reviewer #3: In this study, Jamil et al. analyze the psychological, cultural and academic challenges Saudi Arabian students face while studying in Australia. Of the 219 students included in this analysis, we observe majority respondents to be postgraduates, and well-adjusted to the Australian system and culture.

This revised manuscript is significantly better, and authors have done a good job in responding to majority of prior queries. However, I do have a few minor comments:

1) The manuscript could be significantly improved by a copy editor, or having the document proof read as there are a number of stylistic errors in the text.

Reply: 

The manuscript has been proofread and several sentences rephrased, and stylistic errors have been removed.

Abstract:

2) In the results portion of the abstract, I would suggest authors to mention numerical values instead of writing 'most' or 'few' in lines 3-6 of results. This would help ascertain the actual magnitude of students satisfied with their academic performance and adjustment in Australia.

Reply: 

Thank you for the suggestion. The numerical values have been mentioned as suggested.

3) Last four lines of Results in Abstract "Students' responses regarding responded to a question, "Do you wish to stay in Australia after completion of your studies?", This should be removed to only include the results. Hence authors should instead write “ 77 (35.1%), 119(54.3%), and 23(10.5%) students indicated that they wished to stay in Australia only until completion of their studies, temporarily and permanently, respectively”

Reply: 

Thank you for the suggestion. The sentence has been modified as suggested. 

Introduction

1) Line 4: "The Ministry of Education of the Kingdom of Saudi Arabia (KSA) sponsored 174333 students for abroad education” This should be written as “The Ministry of Education of the Kingdom of Saudi Arabia (KSA) sponsored 174333 students for education abroad”.

Reply: 

Thank you for the suggestion. The sentence has been modified as suggested.

2) As raised earlier, the line “In the case of interactive lectures, the situation becomes rather embarrassing when it becomes difficult for them to take participate in active discussions and question answer- sessions in classes and ward rounds” does not sit so well with me. It may not be embarrassing, rather difficult to partake in fruitful academic discussions due to language barrier, which may lead to lack of participation. I would suggest the authors to revise this sentence.

Reply: 

Thank you for the suggestion. The sentence has been modified as suggested.

3) “International students, especially medical undergraduates, and postgraduates, are among the most successful students in their home countries [6]. This could be because of the better training opportunities and advanced technology available at the universities of the developed countries’ universities.”

This is a bit unclear to me. Do the authors mean developed countries such as Australia or the home country KSA? This should instead be written as international students being successful in their home institutions, who then in search of better training opportunities wish to go abroad. Right now it seems as if the authors are talking about students having better training opportunities in their home country, that’s why they are successful. If this is the case then why would they wish to go abroad? Some clarity would be good to make a strong introduction.

Reply: 

Thank you for the suggestion. The sentence has been modified as suggested.

4) "While in Australia teaching is all mixed in Australia”. Suggest to use the word co-education here

Reply: 

Thank you for the suggestion. The sentence has been modified as suggested.

Results

1) How many Australian institutes was the data obtained from?

Reply:

Data were obtained from 42 Australian institutes (This has been included in the results).

Discussion

1) "The body of the lliterature , indicated” , suggest this to instead write as “current literature indicates”

Reply: 

Thank you for the suggestion. The sentence has been modified as suggested.

2) Future recommendations point 1 "Arabic is a very powerful language, and most of the time, even teacher- student interaction is in the native Arabic language. So, the teachers ensure that , the discussion medium should be only English during classroom discussions and explanations.” Suggest authors to revise this. Difficult to follow the sentence.

Reply: 

Thank you for the suggestion. The sentence has been modified as suggested.

 7. PLOS authors have the option to publish the peer review history of their article (what does this mean?). If published, this will include your full peer review and any attached files.

Do you want your identity to be public for this peer review? For information about this choice, including consent withdrawal, please see our Privacy Policy.

Reviewer #1: No

Reviewer #3: No

---

## [Decision Letter · Decision Letter 2]

1 Nov 2021

PONE-D-21-00081R2

Psychosocial, cultural, and academic challenges to Saudi Arabian students in Australia

PLOS ONE

Dear Dr. Jameel,

Thank you for submitting your manuscript to PLOS ONE. After careful consideration, we feel that it has merit but does not fully meet PLOS ONE’s publication criteria as it currently stands. Therefore, we invite you to submit a revised version of the manuscript that addresses the points raised during the review process.

Please submit your revised manuscript by December 16, 2021. Please include the following items when submitting your revised manuscript:

A 'Response to Reviewers' letter that responds to each point raised by the academic editor and reviewer(s). You should upload this letter as a separate file labeled 'Response to Reviewers'.A marked-up copy of your manuscript that highlights changes made to the original version. You should upload this as a separate file labeled 'Revised Manuscript with Track Changes'.An unmarked version of your revised paper without tracked changes. You should upload this as a separate file labeled 'Manuscript'.

We look forward to receiving your revised manuscript.

Kind regards,

Prof. Ritesh G. Menezes, M.B.B.S., M.D., Diplomate N.B.

Academic Editor

PLOS ONE

Journal Requirements:

Reviewers' comments:

Reviewer's Responses to Questions

**Comments to the Author**

1. If the authors have adequately addressed your comments raised in a previous round of review and you feel that this manuscript is now acceptable for publication, you may indicate that here to bypass the “Comments to the Author” section, enter your conflict of interest statement in the “Confidential to Editor” section, and submit your "Accept" recommendation.

Reviewer #3: All comments have been addressed

Reviewer #4: (No Response)

2. Is the manuscript technically sound, and do the data support the conclusions?

Reviewer #3: Yes

Reviewer #4: Partly

3. Has the statistical analysis been performed appropriately and rigorously? 

Reviewer #3: Yes

Reviewer #4: Yes

4. Have the authors made all data underlying the findings in their manuscript fully available?

Reviewer #3: Yes

Reviewer #4: Yes

5. Is the manuscript presented in an intelligible fashion and written in standard English?

Reviewer #3: Yes

Reviewer #4: Yes

6. Review Comments to the Author

Reviewer #3: (No Response)

Reviewer #4: I had the opportunity to review the revised manuscript, “Psychosocial, cultural, and academic challenges to Saudi Arabian students in Australia”, for possible publication in the Journal of PLOS ONE. The genesis of the presented study is the fact that studying abroad is associated with psychological, cultural, and academic challenges. In their paper, the authors describe the subjective experience of saudi students of medical related professions in Australia. The authors found that saudi students were overall satisfied with their experience in Australia though this is difficult be generalized to other samples or compared with other results. kindly find my few comments enumerated below.

1- The authors stated that the study was conducted during the year of 2019. The authors may specify the exact time-frame over which the study was conducted, and data were collected as the temporal factors may have a significant effect on participants attitudes.

2- Snowball sampling technique was employed to reach out to participants. Could the number of directly contacted participants be specified?

3- “The questionnaire was formulated in English with the help of already published studies”. Could the authors clarify the rationale behind employing this particular questionnaire, and how they insured the validity and the reliability of the questionnaire?

4- Based on the questionnaire responses, how the participants were considered satisfied or not satisfied with their experience? This may be clarified within the methodology section. In other words, when satisfaction was considered as a result?

5- It would be valuable if the Tables and Figures were understandable as stand-alone. I believe that Tables and Figures need more clear descriptive titles.

6- Unmentioned limitation of this study is the use of subjective self-reported measure and associated biases. For instance, acquiescence bias, where people tend to agree on questionnaire, may skew the result form the truth.

7- The conclusions need to be supported by the presented data. The first two sentences in the conclusion section cannot be extrapolated from the presented result.

8- The reference list has some duplicate references. Please omit any duplicate.

7. PLOS authors have the option to publish the peer review history of their article (what does this mean?). If published, this will include your full peer review and any attached files.

Reviewer #3: No

Reviewer #4: No

---

## [Author Response · Author response to Decision Letter 2]

11 Dec 2021

PONE-D-21-00081R2

Psychosocial, cultural, and academic challenges to Saudi Arabian students in Australia

Journal Requirements:

Reply:

We have removed four references because of duplication. We have avoided including any retracted article. 

Reviewers' comments:

Reviewer's Responses to Questions

Comments to the Author

1. If the authors have adequately addressed your comments raised in a previous round of review and you feel that this manuscript is now acceptable for publication, you may indicate that here to bypass the "Comments to the Author" section, enter your conflict of interest statement in the “Confidential to Editor” section, and submit your "Accept" recommendation.

Reviewer #3: All comments have been addressed

Reviewer #4: (No Response)

2. Is the manuscript technically sound, and do the data support the conclusions?

Reviewer #3: Yes

Reviewer #4: Partly

3. Has the statistical analysis been performed appropriately and rigorously?

Reviewer #3: Yes

Reviewer #4: Yes

4. Have the authors made all data underlying the findings in their manuscript fully available?

Reviewer #3: Yes

Reviewer #4: Yes

5. Is the manuscript presented in an intelligible fashion and written in standard English?

Reviewer #3: Yes

Reviewer #4: Yes

6. Review Comments to the Author

Reviewer #3: (No Response)

Reviewer #4: I had the opportunity to review the revised manuscript, “Psychosocial, cultural, and academic challenges to Saudi Arabian students in Australia”, for possible publication in the Journal of PLOS ONE. The genesis of the presented study is the fact that studying abroad is associated with psychological, cultural, and academic challenges. In their paper, the authors describe the subjective experience of saudi students of medical related professions in Australia. The authors found that saudi students were overall satisfied with their experience in Australia though this is difficult be generalized to other samples or compared with other results. kindly find my few comments enumerated below.

1- The authors stated that the study was conducted during the year of 2019. The authors may specify the exact timeframe over which the study was conducted, and data were collected as the temporal factors may have a significant effect on participants attitudes.

Reply: It was conducted from March 15 to June 15, 2019. We have incorporated this in the manuscript.

2- Snowball sampling technique was employed to reach out to participants. Could the number of directly contacted participants be specified?

Reply: We contacted 50 students directly. This statement has been included in the methodology of the revised manuscript.

3- “The questionnaire was formulated in English with the help of already published studies”. Could the authors clarify the rationale behind employing this particular questionnaire, and how they insured the validity and the reliability of the questionnaire?

Reply: We found that these questions thoroughly evaluate the psychosocial, cultural, and academic challenges Saudi Arabian students face in Australia. Therefore, these questions were used in the present study. The content and construct validity were ensured by evaluating these questions by two senior professors and a medical educationist proficient in English. Moreover, the questionnaire was sent to a few students to check its comprehension and later modified after receiving content experts’ and students’ suggestions. The questionnaire reliability was calculated by Cronbach’s alpha, and its value was 0.82. We have included the above information in the revised manuscript.

4- Based on the questionnaire responses, how the participants were considered satisfied or not satisfied with their experience? This may be clarified within the methodology section. In other words, when satisfaction was considered as a result?

Reply: Thank you very much for pointing out this. We have incorporated this point in the methodology. "If more than 60% of students selected options highly satisfactory and satisfactory or unsatisfactory and highly unsatisfactory, were considered satisfied and unsatisfied, respectively.” 

5- It would be valuable if the Tables and Figures were understandable as stand-alone. I believe that Tables and Figures need more clear descriptive titles.

Reply: Thanks for pointing out this. We have modified the titles of all tables and figures.

6- Unmentioned limitation of this study is the use of subjective self-reported measure and associated biases. For instance, acquiescence bias, where people tend to agree on questionnaire, may skew the result form the truth.

Reply: Thank you for pointing out this. We have mentioned a new sentence in the limitation. “A subjective self-reported measure and accompanying biases such as acquiescence and dissent biases are also possible.”

7- The conclusions need to be supported by the presented data. The first two sentences in the conclusion section cannot be extrapolated from the presented result.

Reply: Thank you for pointing out this. We have removed the first two sentences from the conclusion.

8- The reference list has some duplicate references. Please omit any duplicate.

Reply: Thank you for pointing out this. We are sorry for this mistake. All duplicated references have been removed.

7. PLOS authors have the option to publish the peer review history of their article (what does this mean?). If published, this will include your full peer review and any attached files.

Do you want your identity to be public for this peer review? For information about this choice, including consent withdrawal, please see our Privacy Policy.

Reviewer #3: No

Reviewer #4: No

---

## [Decision Letter · Decision Letter 3]

30 Dec 2021

Psychosocial, cultural, and academic challenges to Saudi Arabian students in Australia

PONE-D-21-00081R3

Dear Dr. Jameel,

We’re pleased to inform you that your manuscript has been judged scientifically suitable for publication and will be formally accepted for publication once it meets all outstanding technical requirements.

Kind regards,

Prof. Ritesh G. Menezes, M.B.B.S., M.D., Diplomate N.B.

Academic Editor

PLOS ONE

Reviewers' comments:

Reviewer's Responses to Questions

**Comments to the Author**

1. If the authors have adequately addressed your comments raised in a previous round of review and you feel that this manuscript is now acceptable for publication, you may indicate that here to bypass the “Comments to the Author” section, enter your conflict of interest statement in the “Confidential to Editor” section, and submit your "Accept" recommendation.

Reviewer #4: All comments have been addressed

2. Is the manuscript technically sound, and do the data support the conclusions?

Reviewer #4: Yes

3. Has the statistical analysis been performed appropriately and rigorously? 

Reviewer #4: Yes

4. Have the authors made all data underlying the findings in their manuscript fully available?

Reviewer #4: Yes

5. Is the manuscript presented in an intelligible fashion and written in standard English?

Reviewer #4: Yes

6. Review Comments to the Author

Reviewer #4: (No Response)

7. PLOS authors have the option to publish the peer review history of their article (what does this mean?). If published, this will include your full peer review and any attached files.

Reviewer #4: No

---

## [Editor Report · Acceptance letter]

21 Jan 2022

PONE-D-21-00081R3 

Psychosocial, cultural, and academic challenges to Saudi Arabian students in Australia 

Dear Dr. Jameel:

I'm pleased to inform you that your manuscript has been deemed suitable for publication in PLOS ONE. Congratulations! Your manuscript is now with our production department. 

Kind regards, 

on behalf of

Prof. Dr. Ritesh G. Menezes 

Academic Editor

PLOS ONE